# META KOOPMAN DECOMPOSITION FOR TIME SERIES FORECASTING UNDER DISTRIBUTION SHIFTS

## ABSTRACT

Time series forecasting facilitates various real-world applications and has attracted great research interests. In real-world scenarios, time series forecasting models confront a fundamental issue of temporal distribution shifts, i.e., the statistical properties of time series are evolving over time. In this paper, we utilize Koopman theory to address temporal distribution shifts (TDS). Koopman theory states any time series can be mapped into a Koopman space by proper measurement functions and represented by infinite dimensional linear Koopman operator. Therefore, time series under different distributions can be modeled by different Koopman operators. Considering the linearity of Koopman operators, the Koopman operators for representing time series under different distributions can be decomposed as linear combination of a set of Koopman operators, which we termed as meta Koopman operators. We further theoretically show the infinite dimensional Koopman operators can be approximated by finite matrix multiplications and the meta Koopman operators are equivalent to a set of matrices. Based on the analysis, we propose an auto-encoder framework for implementing the meta Koopman decomposition of time series, which is theoretically able to handle TDS. Extensive experiments conducted on four real-world time series datasets demonstrate the superiority of the proposed model on tackling temporal distribution shifts.

## 1 INTRODUCTION

Time series data are generated in numerous domains including traffic flow Snyder & Do (2019), energy consumption Yu et al. (2016), financial analysis Guen & Thome (2020) and weather condition Zhang et al. (2017). Time series forecasting is one of the most crucial tasks on time series analyzing and accurate forecasting models facilitate various applications in many domains. Great interests have been attracted for building accurate forecasting models, deep learning based models stand out and achieve state-of-the-art forecasting accuracy Zhou et al. (2021); Lee et al. (2022); Li et al. (2019). As the world keeps evolving, the statistical properties of time series can change over time, such phenomena is termed as temporal distribution shifts (TDS). Recently, increasing efforts have been made for building more robust and accurate deep learning models for time series data under distribution shifts Arik et al. (2022); Liu et al. (2022); Masserano et al.; Kim et al., which can be divided into two categories, data-orient methods and feature-orient methods.

Data-orient methods Passalis et al. (2019); Kim et al.; Liu et al. (2022) try to alleviate the distribution variation by normalizing statistical properties of input data. For instance, RevIN Kim et al. proposes a reversible instance normalization which normalizes the input into distributions with means of $0$ and variances of $1$ for processing and denormalizes the forecasts back to original scale. Although the normalized data are constrained with same statistical properties, e.g., mean and variance, the distributions of normalized data are still diverse since we can not determine a distribution only according to mean and variance.

Feature-orient methods Du et al. (2021); Woo et al. (2022); Arik et al. (2022) propose model architectures or learning strategies for mining generalizable features which are expected to represent time series under various distributions. AdaRNN Du et al. (2021) characterizes different distributions among time series data and extracts invariant features among different distributions. However, mining invariant features under-utilizes the diversity of distribution in the time series, which induces low representative capacity of AdaRNN. Woo et al. (2022) further proposes to extract disentangled seasonal-trend features for better representing time series segments from different distributions. While seasonal and trend features are commonly utilized in series analysis, seasonal-trend features could be insufficient for modeling complex distribution shifts.

The key assumption of recent feature-orient methods is that the time series of interest, both the training and testing parts, consists of a set of meta distributions, which can be fully extracted from training data and are able to compose shifted distribution among testing data. Nevertheless, how to effectively capture the meta distributions and model the distribution shifts remains challenging and an open problem.

In this paper, following the assumption of existing works, we apply Koopman theory Koopman (1931) to address the issue of temporal distribution shifts. Koopman theory states that any dynamics, including time series in our case, can be mapped into a Koopman space by proper measurement functions and represented by linear Koopman operators on the space. Therefore, time series under different distributions can be modeled by different Koopman operators. Considering the linearity of Koopman operators, the Koopman operators for representing time series under different distributions can be decomposed as linear combination of a set of Koopman operators, which we termed as meta Koopman operators. We further theoretically show the infinite dimensional Koopman operators can be approximated by finite matrix multiplications and the meta Koopman operators are equivalent to a set of matrices. By introducing Koopman theory, representing time series under distribution shifts is equivalent to constructing distribution-specific Koopman operators based on the meta Koopman operators. Based on the above analysis, we propose an auto-encoder framework for implementing the meta Koopman decomposition of time series. Specifically, a temporal trend aware encoder is proposed to generate measurements of time series states, which can be modeled by a linear Koopman operator. Based on the measurements of historical states, a novel meta Koopman operators matching mechanism is proposed to construct the Koopman operator by combining a set of learnable matrices termed as meta Koopman operators. The combination of meta operators is dynamic and data-driven, which endows the proposed framework with the ability of modeling dynamic temporal distributions, i.e., temporal distribution shifts. Then the decoder of our model makes predictions based on the constructed Koopman operator and measurements of historical states.

Our contributions are summarized as,

- We analyze the feasibility of utilizing Koopman theory to address temporal distribution shifts and propose a meta Koopman operators matching module to construct proper Koopman operators by linearly combining meta Koopman operators for modeling time series under different distribution.

- To implement the meta Koopman decomposition, we propose an auto-encoder framework which generates dynamic data-driven measurements of time series and recover time series based on the measurements.

- Extensive experiments conducted on four real-world time series datasets demonstrate the superiority of the proposed model on tackling temporal distribution shifts.

## 2 PRELIMINARY

### 2.1 TIME SERIES FORECASTING

We first formally define the problem of time series forecasting. Time series data can be denoted as a set of observations $\{x_t \in \mathbb{R}^d\}$ of a dynamical system states, where $d$ is the dimension of states and t denotes

discrete time steps. The goal of time series forecasting is to find a function $f$ to forecasting future $q$-step states based on historical $p$-step states as,

$$[x_{t+1}, x_{t+2}, \cdots, x_{t+q}] = f([x_t, x_{t-1}, \cdots, x_{t-p+1}]) \tag{1}$$

## 2.2 KOOPMAN THEORY FOR TIME SERIES FORECASTING

As in Azencot et al. (2020), the time series of interest can be described by a discrete-time evolution function as,

$$x_{t+1} = F(x_t) \tag{2}$$

where $F(x_t)$ updates the states of time series from time $t$ to $t + 1$ on a finite dimensional manifold $\mathcal{X} \subset \mathbb{R}^d$. Koopman theory Koopman (1931) suggests that any such kind of nonlinear dynamics can be transformed into a Koopman space where the evolution of states are linear. Formally, for time series in Eq.2, there exists a linear infinite dimensional Koopman operator $\mathcal{K} : \mathcal{G}(\mathcal{X}) \to \mathcal{G}(\mathcal{X})$ so that

$$\mathcal{K}g(x_t) = g(F(x_t)) = g(x_{t+1}) \tag{3}$$

where $\mathcal{G}(\mathcal{X})$ is a set of measurement functions $g : \mathcal{X} \to \mathbb{R}$. Therefore, making one-step prediction with Koopman operator $\mathcal{K}$ and measurement function $g(x_t)$ can be achieved by,

$$x_{t+1} = \Psi(g(x_{t+1})) = \Psi(\mathcal{K}g(x_t)) \tag{4}$$

where $\Psi$ is a function to reconstruct time series states according the measurements in Koopman space. Considering $g(x)$ reduces the dimension of states $x_t$, we may have multiple measurement functions $\mathbf{g} = [g_1, g_2, \cdots, g_M]^T$ to maintain sufficient information for such reconstruction.

Finding proper Koopman operator can be intractable, we next show one can avoid finding such operator and achieve Eq.3 by infinite dimensional matrix multiplications.

Since $\mathcal{K}$ is a linear operator on function space $\mathcal{G}$, $\mathcal{K}$ has a infinite set of eigenfunctions $\Phi = \{\varphi_k : \mathcal{X} \to \mathbb{R}\}$. An eigenfunction $\varphi_k$ of $\mathcal{K}$ satisfies,

$$\mathcal{K}\varphi_k(x_t) = \lambda_k \varphi_k(x_t) = \varphi_k(x_{t+1}) \tag{5}$$

where $\lambda_k$ is the corresponding eigenvalue of eigenfunction $\varphi_k$. And each of the individual measurements $g_i$ in $\mathbf{g}$ may be expanded in terms of a basis of eigenfunctions,

$$g_i(x) = \sum_{j=1}^{\infty} v_{ij}\varphi_j(x) \tag{6}$$

where $\mathbf{v}_i = [v_{i1}, v_{i2}, \cdots]$ is the mode of $g_i$ in Koopman space. Further, for $\mathbf{g} = [g_1, g_2, \cdots, g_M]^T$, we have,

$$x_{t+1} = \Psi(\mathcal{K}\mathbf{g}(x_t)) = \Psi(K\Phi(x_t)) \tag{7}$$

where $\mathbf{g}(x_t) \in \mathbb{R}^M$ denotes the $M$-dimension measurements generated by $M$ measurement functions $K \in \mathbb{R}^{M \times \infty}$ is defined as $K_{ij} = \lambda_j \mathbf{v}_{ij}$. So far, the Koopman operator is converted to a matrix multiplication in the Koopman space spanned by eigenfunctions $\Phi$. While ensuring clarity of expression, $K$ will also be denoted as Koopman operator in the following.

## 3 METHODOLOGY

According to Eq.7, we propose an auto-encoder framework to implement Koopman theory for tackling time series forecasting under distribution shifts, as shown in Fig.1. Noting that both the Koopman operator $K$ and the set of eigenfunctions $\Phi$ in Eq.7 are infinite dimensional, we propose a finite dimensional approximation

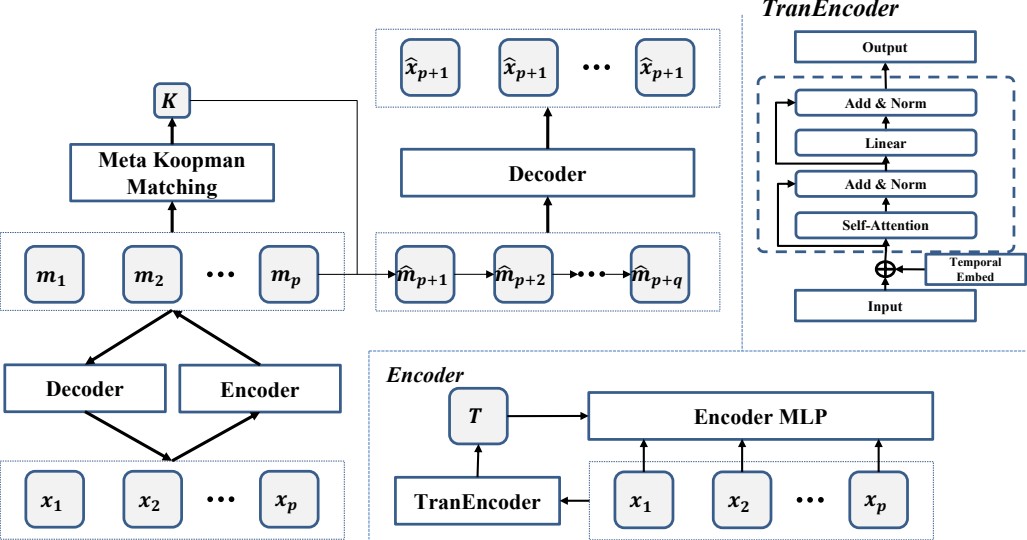

Figure 1: Architecture of the proposed framework.

of both in our framework. Specifically, the encoder works as finite subset of eigenfunctions and transform time series into Koopman space. Then a meta Koopman operators matching module is proposed to construct proper Koopman operator for samples from diverse distributions. With the constructed Koopman operator, the measurements at future time steps can be estimated by applying Eq.7. Finally, the decoder generates predictions based on the estimated measurements at future time steps.

### 3.1 DYNAMIC DATA-DRIVEN ENCODER FOR MEASUREMENTS

As mentioned, the set of eigenfunctions $\Phi$ in Eq.7 are infinite dimensional, a finite dimensional approximation of $\Phi$ is required. In fact, not all of the infinite eigenfunctions have to be involved in spanning Koopman space, which favors our finite approximation of eigenfunctions. For instance, considering an originally linear time series, only an identical mapping need to be involved for generating measurements without any loss of representative capacity. Therefore, with proper approximating strategy of the combination of infinite eigenfunctions, the loss of representative capacity of such approximation is acceptable. Taking both representative capacity and temporal distribution shifts into consideration, we argue that a proper approximation should 1) generate diverse measurements of time series states for ensuring representative capacity; 2) be dynamic and adaptive to local distributions for addressing distribution shifts. To this end, we propose a dynamic data-driven encoder to transform input time series into Koopman space.

Given historical time series states $X = [x_1, x_2, \cdots, x_p] \in \mathbb{R}^{p \times d}$, we first employ a transformer encoder layer and the positional encoding module in Vaswani et al. (2017) to capture the temporal trend $T$ of $X$.

$$attn = \text{TranEncoder}(\text{PosEncoding}(X)) \in \mathbb{R}^{p \times d_t}$$

$$T = \sum_{i=1}^{p} w_i attn_{i,:} \tag{8}$$

where $\{w_i \in \mathbb{R} | i \in [1, p]\}$ are learnable weights and $d_t$ is a hyperparameter controlling the dimension of $T$. The trend $T \in \mathbb{R}^{d_t}$ is weighted sum of the output of transformer encoder across temporal dimension. The

trend $T$ encodes local distribution information of $X$, and is further used for generating measurements of $X$ and reconstructing time series states based on the measurements.

$T$ is then concatenated with $X$ and the concatenation is denoted as $Z = [z_1, z_2, \cdots, z_p]$, where $z_i = x_i || T, z_i \in \mathbb{R}^{d+d_t}$. An $L$-layer MLP is applied on the feature dimension of $Z$ for generating measurements $M$ of $X$.

$$M = \text{MLP}(Z) = [\text{MLP}(z_1), \text{MLP}(z_2), \cdots, \text{MLP}(z_p)] \in \mathbb{R}^{p \times d_m} \tag{9}$$

$d_m$ is the dimension of measurements $M$. Since the dimension of measurements is reduced from infinite in Eq.7 to $d_m$ here, we further suggest to maximize the diversity of different measurements for ensuring representative capacity as,

$$Loss_{div} = -\frac{2}{d_m(d_m - 1)} \sum_{1 \leq i < j \leq d_m} \frac{< M_{:,i}, M_{:,j} >}{||M_{:,i}|| ||M_{:,j}||} \tag{10}$$

where $< M_{:,i}, M_{:,j} >$ means inner product of $M_{:,i}$ and $M_{:,j}$. Eq.10 makes the measurements uniformly distributed in the measurement space.

## 3.2 META KOOPMAN OPERATORS MATCHING

According to Eq.7, benefiting from the linearity of Koopman operator, a Koopman operator can be decomposed as linear combination of operators. Therefore, we can actually combine various Koopman operators to model different distributions of time series and thus address the issue of temporal distribution shifts. To this end, we propose a meta Koopman operators matching module to implement meta Koopman decomposition.

Specifically, we maintain a set of learnable meta Koopman operators $\mathcal{M} = [K_1, K_2, \cdots, K_k]$, where $K_i \in \mathbb{R}^{d_m \times d_m}$ denotes a learnable Koopman operator and $k$ is the number of meta Koopman operators. Given measurements $M = [m_1, m_2, \cdots, m_p]$ of $p$ historical time steps, the goal of meta Koopman operators matching module is to find a linear combination of meta Koopman operators to best model the dynamics of $M$ as,

$$\min_{\lambda} \text{mean}(|\sum_{i=1}^{k} \lambda_i K_i M_1 - M_2|) \tag{11}$$

where $M_1 = [m_1, m_2, \cdots, m_{p-1}]$ and $M_2 = [m_2, m_3, \cdots, m_p]$. This optimization goal indicates that the constructed Koopman operator $K = \sum_{i=1}^{k} \lambda_i K_i$ have to match the dynamics on historical data. Solving this optimization problem is intractable and time-consuming, so we propose a similarity based matching mechanism as,

$$\lambda_i = \frac{\exp(K_i M_1 - M_2)}{\sum_{j=1}^{k} \exp(K_j M_1 - M_2)} \tag{12}$$

where the meta Koopman operators are combined according to the similarity between the dynamics they determine and the dynamics of $M$.

## 3.3 FORECASTING AND LOSS FUNCTION

Given measurements $M = [m_1, m_2, \cdots, m_p]$ of $p$ historical time steps and constructed Koopman operator $K$, making prediction to measurements $\hat{M}' = [\hat{m}_{p+1}, \hat{m}_{p+2}, \cdots, \hat{m}_{p+q}]$ of future $q$ time steps is rather simple by matrix multiplication of measurements and Koopman operator as,

$$\hat{m}_{p+i} = K^i m_p \tag{13}$$

and the final prediction $\hat{X}' = [\hat{x}_{p+1}, \hat{x}_{p+1}, \cdots, \hat{x}_{p+q}]$ can be made by applying decoder $\Psi$ on $\hat{M}$ as,

$$\hat{x}_{p+i} = \Psi(\hat{m}_{p+i}) \tag{14}$$

where the decoder $\Psi$ has similar architecture to the encoder, i.e., $\Psi$ is also a $L$-layer MLP. Then, a supervised forecasting loss can be obtained,

$$Loss_{pre} = \text{MAE}(X', \hat{X}') = \frac{1}{qd} \sum_{i=p+1}^{p+q} \sum_{j=1}^{d} |x_{i,j} - \hat{x}_{i,j}| \tag{15}$$

as shown, we apply mean absolute error (MAE) as supervised loss. Also, since the proposed model has an auto-encoder architecture, a reconstruction loss is introduced for training the encoder and decoder,

$$Loss_{rec} = \text{MAE}(X, \Psi(\Phi(X))) \tag{16}$$

The final loss of our framework is weighted sum of the three losses, $Loss_{div}$, $Loss_{pre}$ and $Loss_{rec}$,

$$Loss = \alpha_1 Loss_{div} + \alpha_2 Loss_{rec} + Loss_{pre} \tag{17}$$

where both $\alpha_1$ and $\alpha_2$ are hyperparameters for tuning the weights.

## 4 EXPERIMENTS

### 4.1 DATASETS

The proposed method is evaluated on four time series datasets: Crypto [1], Weather [2], Electricity [3] and Traffic [4]. Table.1 summarizes useful statistics of the four datasets. Crypto dataset contains 8 kinds of trade features for 14 cryptocurrencies. The data are collected minutely and there are 1.9 million time steps in this dataset. Electricity dataset contains the electricity consumption of 321 clients, which is collected hourly. There are 26 thousand time steps in Electricity dataset. Weather dataset contains 21 meteorological indicators for a range of 1 year in Germany, which are recorded every 10 minutes. There are 52

Table 1: Datasets statistics.

| Datasets | Frequency | Length | Features |
|---|---|---|---|
| Crypto | 1 Minute | 1.9 million | 112 |
| Weather | 10 Minutes | 52695 | 21 |
| Electricity | 1 Hour | 26304 | 321 |
| Traffic | 1 Hour | 17544 | 862 |

thousand time steps in this dataset. Traffic dataset contains the occupation rate of freeway system measured hourly by 862 sensors across California. There are 17 thousand time steps in Traffic dataset. As shown in Fig.2, Crypto and Weather have more complex temporal patterns. These two datasets are suitable for evaluating the performance of proposed model on handling temporal distribution shifts. The other two datasets are selected to evaluate the proposed model on canonical settings.

### 4.2 EXPERIMENTAL SETTINGS AND BASELINES

**Experimental settings.** For fair comparison, we follow the data processing in Zhou et al. (2021) on Electricity, Traffic and Weather. All the three datasets are split into training set, validation set and test set with ratio of $7 : 1 : 2$. The input length $p$ is fixed to 96 and the prediction lengths are set to 96, 192, 336 and 720, respectively. The original task of Crypto is to predict 3-step future states using 15-step historical states. We keep the size of historical window and set the prediction steps to 3, 6, 12 and 15, respectively. Similarly, Crypto is also split into training set, validation set and test set with ratio of $7 : 1 : 2$. All datasets are zero-mean

---

[1] https://www.kaggle.com/c/g-research-crypto-forecasting/

[2] https://www.bgc-jena.mpg.de/wetter/

[3] https://archive.ics.uci.edu/ml/datasets/ElectricityLoadDiagrams

[4] http://pems.dot.ca.gov

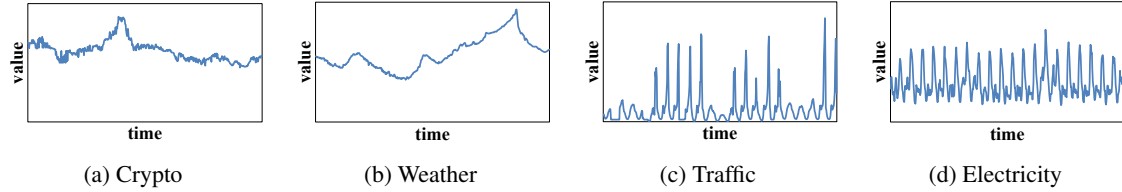

|  (a) Crypto | (b) Weather | (c) Traffic | (d) Electricity |

Figure 2: Visualization of samples from different datasets. Crypto and Weather suffer severer temporal distribution shifts.

Table 2: Forecasting performance on datasets without temporal distribution shifts.

| Model | Metrics | Electricity | | | | Traffic | | | |
|---|---|---|---|---|---|---|---|---|---|
|  |  | 96 | 192 | 336 | 720 | 96 | 192 | 336 | 720 |
| LSTNet | MSE | 0.680 | 0.725 | 0.828 | 0.957 | 1.107 | 1.157 | 1.216 | 1.481 |
|  | MAE | 0.645 | 0.676 | 0.727 | 0.811 | 0.685 | 0.706 | 0.730 | 0.805 |
| Reformer | MSE | 0.312 | 0.348 | 0.350 | 0.340 | 0.732 | 0.733 | 0.742 | 0.755 |
|  | MAE | 0.402 | 0.433 | 0.433 | 0.420 | 0.423 | 0.420 | 0.420 | 0.423 |
| LogTrans | MSE | 0.258 | 0.266 | 0.280 | 0.283 | 0.684 | 0.685 | 0.733 | 0.717 |
|  | MAE | 0.357 | 0.368 | 0.380 | 0.376 | 0.384 | 0.390 | 0.408 | 0.396 |
| Informer | MSE | 0.274 | 0.296 | 0.300 | 0.373 | 0.719 | 0.696 | 0.777 | 0.864 |
|  | MAE | 0.368 | 0.386 | 0.394 | 0.439 | 0.391 | 0.379 | 0.420 | 0.472 |
| Pyraformer | MSE | 0.498 | 0.828 | 1.476 | 4.090 | 0.684 | 0.692 | 0.699 | 0.712 |
|  | MAE | 0.299 | 0.312 | 0.326 | 0.372 | 0.393 | 0.394 | 0.396 | 0.404 |
| Autoformer | MSE | 0.201 | 0.222 | 0.231 | 0.254 | 0.613 | 0.616 | 0.622 | 0.660 |
|  | MAE | 0.317 | 0.334 | 0.338 | 0.361 | 0.388 | 0.382 | 0.337 | 0.408 |
| Fedformer | MSE | 0.183 | 0.195 | 0.212 | 0.231 | 0.562 | **0.562** | **0.570** | **0.596** |
|  | MAE | 0.297 | 0.308 | 0.313 | 0.343 | 0.349 | 0.346 | **0.323** | **0.368** |
| Ours | MSE | **0.168** | **0.181** | **0.199** | **0.220** | **0.561** | 0.581 | 0.620 | 0.663 |
|  | MAE | **0.271** | **0.287** | **0.301** | **0.318** | **0.339** | **0.345** | 0.331 | 0.369 |

normalized. Two metrics, MAE and MSE, are employed for evaluation. The proposed model is implemented in Python with PyTorch 1.9, trained and tested with one Nvidia Tesla V100 16GB. We utilize Adam for tuning the parameters with the maximum epochs of 100 with initial learning rate as 0.001 on Electricity, 0.003 on Traffic, 0.005 on Weather and 0.005 on Crypto. The learning rate decays to 1% of its initial value when the loss on validation set does not improve for 15 epochs. To achieve better performance, we apply different settings on different datasets which are chosen through a carefully parameter-tuning process on the validation set.

**Baselines.** We compare our model with different baselines on different datasets. Electricity and Traffic suffer few temporal distribution shifts, and we employ several canonical time series forecasting methods, which achieve state-of-the-art performance on Electricity and Traffic, including, 1) LSTNet Lai et al. (2018) proposed a deep learning framework to discover long-term patterns for time series trends. 2) Reformer Kitaev et al. (2020) introduces a local-sensitive hashing for reducing the complexity. 3) LogTrans Li et al. (2019) also focuses on reducing the time complexity of vanilla attention and proposes a log-sparse attention. 4)

Table 3: Forecasting performance on datasets with temporal distribution shifts.

| Model | Metrics | Weather | | | | Crypto | | | |
|---|---|---|---|---|---|---|---|---|---|
| | | 96 | 192 | 336 | 720 | 3 | 6 | 12 | 15 |
| LogTrans | MSE | 0.458 | 0.658 | 0.797 | 0.869 | 0.0070 | 0.0076 | 0.0082 | 0.0074 |
| | MAE | 0.490 | 0.589 | 0.652 | 0.675 | 0.0038 | 0.0038 | 0.0041 | 0.0038 |
| Reformer | MSE | 0.689 | 0.752 | 0.639 | 1.130 | 0.0105 | 0.0087 | 0.0065 | 0.0096 |
| | MAE | 0.596 | 0.638 | 0.596 | 0.792 | 0.0046 | 0.0041 | 0.0037 | 0.0044 |
| Informer | MSE | 0.300 | 0.598 | 0.578 | 1.059 | 0.0046 | 0.0069 | 0.0059 | 0.0090 |
| | MAE | 0.384 | 0.544 | 0.523 | 0.741 | 0.0030 | 0.0035 | 0.0033 | 0.0041 |
| Pyraformer | MSE | 0.354 | 0.673 | 0.634 | 0.942 | 0.0054 | 0.0078 | 0.0065 | 0.0080 |
| | MAE | 0.392 | 0.597 | 0.592 | 0.723 | 0.0030 | 0.0038 | 0.0037 | 0.0040 |
| Autoformer | MSE | 0.266 | 0.307 | 0.359 | 0.419 | 0.0040 | 0.0035 | 0.0037 | 0.0036 |
| | MAE | 0.336 | 0.367 | 0.395 | 0.428 | 0.0026 | 0.0024 | 0.0025 | 0.0024 |
| AdaRNN | MSE | 0.283 | 0.328 | 0.393 | 0.458 | 0.0043 | 0.0043 | 0.0044 | 0.0042 |
| | MAE | 0.366 | 0.394 | 0.434 | 0.481 | 0.0028 | 0.0030 | 0.0031 | 0.0031 |
| HyperGRU | MSE | 0.202 | 0.278 | 0.352 | 0.441 | 0.0031 | 0.0032 | 0.0036 | 0.0037 |
| | MAE | 0.315 | 0.337 | 0.385 | 0.473 | 0.0024 | 0.0022 | 0.0024 | 0.0026 |
| Ours | MSE | **0.171** | **0.243** | **0.322** | **0.412** | **0.0026** | **0.0028** | **0.0033** | **0.0035** |
| | MAE | **0.220** | **0.281** | **0.334** | **0.411** | **0.0017** | **0.0018** | **0.0021** | **0.0023** |

Informer Zhou et al. (2021) selects top-k in attention matrix with a KL-divergence based method. 5) Pyraformer Liu et al. (2021) explores the multi-resolution representation of the time series and utilizes the multi-resolution features to generate more accurate forecasting. 6) Autoformer Wu et al. (2021) proposes a novel auto-correlation module to replace the vanilla self attention block. 7) Fedformer Zhou et al. (2022) is a state-of-art transformer-based time series forecasting model, which utilizes frequency information to enhance transformer.

Considering the severe temporal distribution shifts in Weather and Crypto, we additionally include several time series forecasting models designed for addressing distribution shifts, including, 1) AdaRNN Du et al. (2021) characterizes temporal distribution and learn distribution invariant representations for robustness and generalization. 2) HyperGRU Duan et al. (2023) proposes to dynamically generate parameters for its main layers to make accurate predictions. The results of all baselines are either reproduces with public available code or cited from existing papers.

### 4.3 RESULTS ON ELECTRICITY AND TRAFFIC

Table.2 shows the comparison of our model with baselines on Electricity and Traffic. As demonstrated, our model achieves the best performance accuracy on Electricity and outperforms the best baseline Fedformer with average increments of 6.56% and 6.67% on MSE and MAE respectively. However, we find that our model fails to outperform Fedformer on Traffic on long-term forecasting. Considering our model generates predictions in an auto-regressive manner, a performance drop on long-term forecasting is foreseeable. We further argue that such failure also results from the unsatisfying ability of our model to handle higher-dimensional features in Traffic, which will be further explored in later section. Although our model fails to achieve the best performance on all settings on Traffic, the forecasting accuracy of our model is acceptable and satisfying. The performance of our model on Electricity and Traffic validates the effectiveness of our model on canonical time series forecasting.

## 4.4 RESULTS ON CRYPTO AND WEATHER

In Table.3, we show the forecasting accuracy of baselines and our model on Crypto and Weather with respect to MSE and MAE. Since Crypto and Weather suffer severe temporal distribution shifts, the performance on Crypto and Weather demonstrate the ability of models to handle temporal distribution shifts. As shown, the proposed model achieves the best performance and outperform HyperGRU with average margins of $[10.7\%, 18.3\%]$ and $[11.8\%, 18.9\%]$ with respect to MSE and MAE on Weather and Crypto respectively. The performance improvement of our model on datasets with temporal distribution shifts are rather huge, which validates the superiority of our model on tackling distribution shifts in time series.

## 4.5 ABLATION STUDY

In this part, we evaluate the contribution of key components of our model to the forecasting performance on Crypto, since Crypto suffers severe temporal distribution shifts. Concretely, the key components of our model are: 1) trend-aware measurements in encoder, as measurement functions are essential for Koopman theory; 2) measurement diversity ensuring loss in Eq.10, which is expected to ensure the representative capacity of measurements; 3) the meta Koopman operators matching module, which models the dynamics of measurements of time series states. Therefore, we design a series of variants of our model, 1) **w/o trend** removes the temporal trend in Eq.8 from origin model. 2) **w/o div** removes $Loss_{div}$ from the final loss in Eq.17. 3) **w/o match** removes meta Koopman operators matching mechanism and defines a learnable matrix. Table.4 shows the performance comparison between the variants and origin model. As can be found, each component contributes to the superiority of our model on tackling temporal distribution shifts.

Table 4: Forecasting performance of different variants on Crypto.

| Variants | Metrics | 12 | 15 |
|---|---|---|---|
| w/o trend | MSE | 0.0051 | 0.0054 |
| | MAE | 0.0046 | 0.0050 |
| w/o div | MSE | 0.0042 | 0.0044 |
| | MAE | 0.0037 | 0.0039 |
| w/o match | MSE | 0.0053 | 0.0058 |
| | MAE | 0.0048 | 0.0051 |
| origin | MSE | 0.0033 | 0.0035 |
| | MAE | 0.0021 | 0.0023 |

## 5 CONCLUSION

In this paper, a time series forecasting model combining with Koopman theory is proposed to address the issue of temporal distribution shifts. The proposed model has an auto-encoder architecture. The encoder works as measurement functions to map time series into measurements so that the complex dynamics of time series can be modeled by applying linear infinite dimensional Koopman operators on the measurements. The decoder generates predictions of future states according to estimated measurements of future states. A meta Koopman operators matching mechanism is designed to generate proper matrices to approximate Koopman operator to model the dynamics of time series under different temporal distributions. Extensive experiments on four real-world datasets validate the superiority of the proposed model. Meanwhile, some limitations are found during experiments. The model follows an auto-regressive manner to generate predictions and is thus risky to suffer error accumulation. Also, mapping time series under diverse distributions into a linear space requires high dimensional measurements, leading to the compromise between efficiency and performance.

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

## A    MORE IMPLEMENTATION DETAILS

In this section, we further detail the implementation of our model. As mentioned, MAE and MSE are used for evaluating our model and baselines. Given ground truth $\mathbf{X}$ and $\hat{\mathbf{X}}$, the definitions of the two metrics are as below,

$$\text{MAE}(\mathbf{X}, \hat{\mathbf{X}}) = mean(sum(|\mathbf{X} - \hat{\mathbf{X}}|))$$
$$\text{MSE}(\mathbf{X}, \hat{\mathbf{X}}) = mean(sum((\mathbf{X} - \hat{\mathbf{X}})^2)) \tag{18}$$

The initial learning rate of all datasets are selected from $1e-2$ to $1e-4$. The initial values of $\alpha_1$ and $\alpha_2$ range from $0.01$ to $1$, and the best $\alpha_1$ and $\alpha_2$ are $0.3$ and $0.1$ respectively. The dimension of measurements is set to $10$ times of dimension of features of input time series. The number of layers $L$ of measurement MLP ranges from $1$ to $10$, and the best $L$ is $4$.

## B    MULTI-STEP PREDICTION ON CRYPTO

We present detailed prediction accuracy on each time step in Fig.3. As shown, compared with baselines, the proposed model achieves lower MSE and MAE on all steps. Such result indicates the proposed model has stable forecasting performance on multi-step forecasting setting.

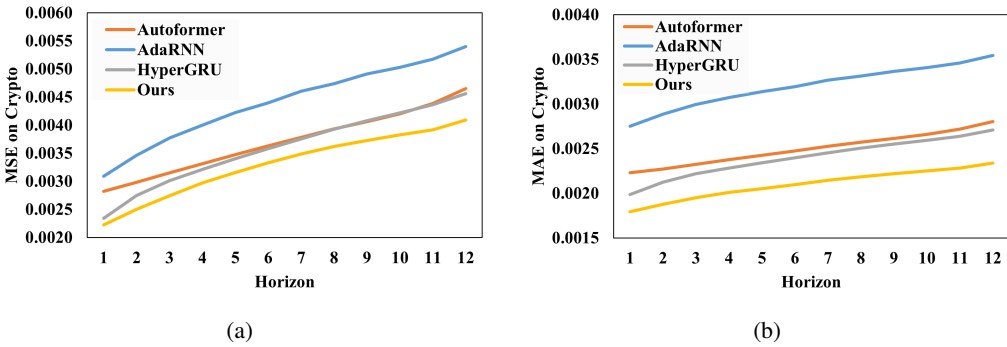

(a)                                                                  (b)

Figure 3: Multi-step prediction comparison on Crypto.

### B.0.1    IMPACT OF DIMENSION OF STATES

As mentioned, the proposed model fails to outperform the best baseline on Traffic on long-term forecasting. We argue such failure results from the unsatisfying ability of our model to handle higher-dimensional features. To expore the impact of dimension of time series states, we randomly select $100$ to $800$ states from Traffic, and compare the performance of our model on the subsets. $96$ historical states are used to predict next $96$-step states. As illustrated in Fig.4, when the dimension of states increases, the performance of our model drops more than Fedformer, which indicates the limitation of our model on handling huge states dimension. The reason of such limitation is that, according to Eq.9, we approximate infinite dimensional measurements in Koopman space with finite dimensional measurements. When the dimension of time series states increases, the dimension required for building reliable linear measurements increases. However, in consideration of time and space complexity, the dimension of measurements can not be very large. Therefore, when the dimension of time series states increases, there must be a compromise between model complexity and performance.

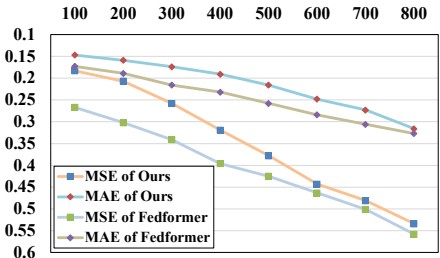

Figure 4: Performance on Traffic with diverse states dimension.

## C  VISUALIZATIONS ON CRYPTO

To more intuitively validate the effectiveness of our model, we further visualize some prediction samples of our model on Crypto.

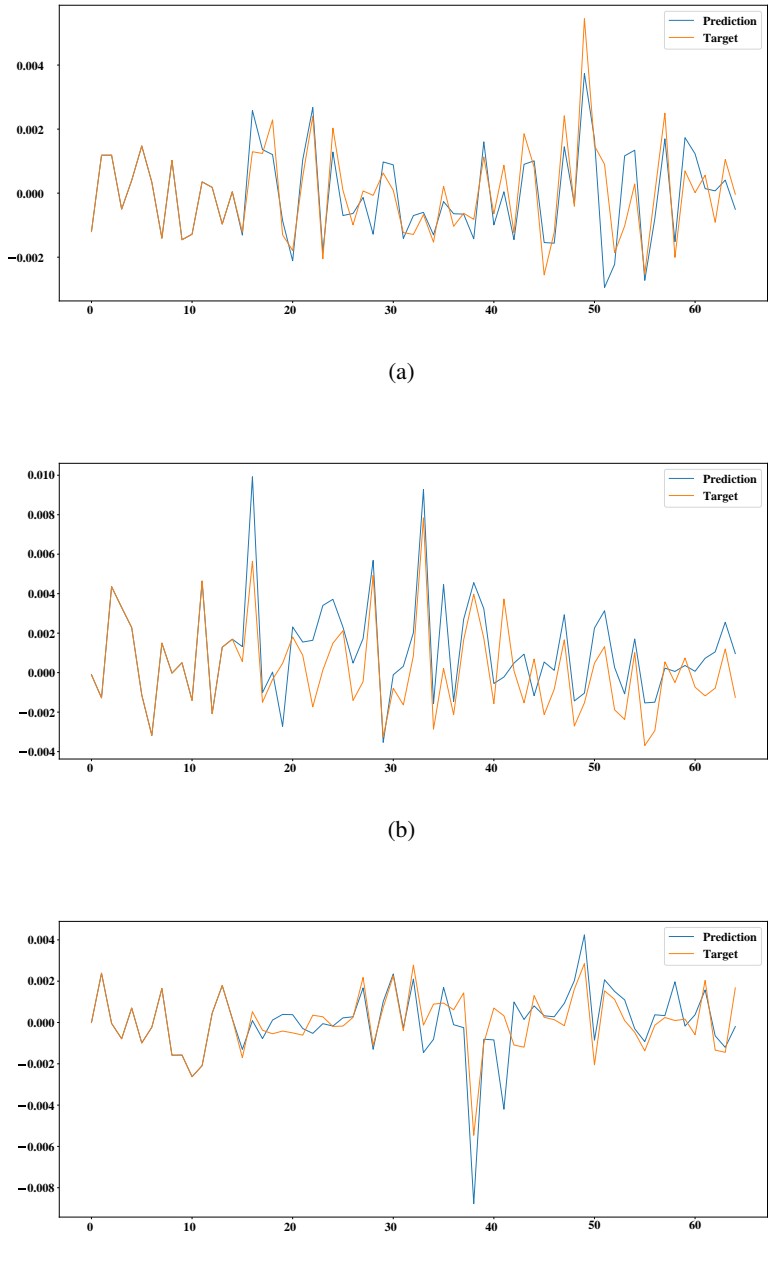

(a)

(b)

(c)

Figure 5: Visualizations of predictions on Crypto.

