# OpenReview forum: "Meta Koopman Decomposition for Time Series Forecasting Under Distribution Shifts"
_ICLR.cc/2024/Conference — ICLR 2024 Conference Withdrawn Submission_

### Official Review · Reviewer_wJjt · 2023-10-18

**Soundness:** 3 good
**Presentation:** 2 fair
**Contribution:** 2 fair
**Rating:** 5
**Confidence:** 3

**Summary:**

This paper proposes a methodology that applies Koopman theory to solve the Temporal Distribution Shift (TDS), which is prominent in the time series field. Through Koopman theory, time series of different distributions can be sent to Koopman space through an appropriate measurement function, and this Koopman space can be expressed as linear Koopman operators. Considering Koopman's linearity and Koopman operator, different time series distributions can be converted to Koopman operators. It can be decomposed into linear combinations. This is called the meta-koopman operator. In particular, it is said that the TDS problem can be eliminated because it linearly combines meta Koopman operators for time series modeling in different distributions.

**Strengths:**

1. Solving the time series distribution shift problem is an important problem in time series prediction, and the attempt to solve the time-series distribution shift by applying Koopman theory seems novel and good.
2. Koopman theory The theory was unfamiliar to me, but it is well explained in the paper.

**Weaknesses:**

1. There is a lack of experimentation compared to other long-term time series forecasting models. I would like to see the results of the experiment on the ETTdataset or National Illness dataset.
2. Please compare with LTSF-Linear[1], which solves the time-series distribution shift problem.
3. We can only know from the prediction experiment results that Koopman theory has solved the distribution shift. Can't you show visualization results or ablation study results showing that Koopman theory solves distribution shift?
4. What is difference with [2]?
5. Please compare with other forecasting models based on Koopman theory such as [2].


[1] Zeng et al., Are Transformers Effective for Time Series Forecasting?, AAAI 2023
[2] Wang et al., KOOPMAN NEURAL FORECASTER FOR TIME SERIES WITH TEMPORAL DISTRIBUTION SHIFTS, ICLR 2023

**Questions:**

1. There is a lack of experimentation compared to other long-term time series forecasting models. I would like to see the results of the experiment on the ETTdataset or National Illness dataset.
2. Please compare with LTSF-Linear[1], which solves the time-series distribution shift problem.
3. We can only know from the prediction experiment results that Koopman theory has solved the distribution shift. Can't you show visualization results or ablation study results showing that Koopman theory solves distribution shift?

---

### Official Review · Reviewer_anBH · 2023-10-26

**Soundness:** 2 fair
**Presentation:** 2 fair
**Contribution:** 2 fair
**Rating:** 3
**Confidence:** 5

**Summary:**

This paper addresses the task of forecasting time series data in the presence of distribution shifts. The authors introduce a novel approach that leverages diverse Koopman operators to capture the distinct distributions within a time series, then decomposes the Koopman operator into a linear combination of several Koopman operators. To facilitate Koopman learning, an autoencoder framework is employed. The effectiveness of this method is assessed through rigorous evaluation using four real-world time series datasets.

**Strengths:**

1. The idea of leverage Koopman operator to tackle time series distribution shift is interesting and novel.
2. The paper is written in a clear manner for the reader to follow.

**Weaknesses:**

1. A significant drawback of this paper is the absence of a literature review on recent developments in the field of deep Koopman methods, despite the method's foundation in Koopman theory. Numerous works have emerged in recent years, harnessing Koopman theory for time-series analysis, as exemplified by [1-4].

2. Another point of concern is the lack of a comparative analysis with existing Koopman-based methods for time series forecasting, as indicated by [2-4]. This omission detracts from the persuasiveness of the evaluation section.

I recommend that the authors undertake a comprehensive review of recent advancements in deep Koopman methods and carefully design the experimental part to enable a fair and meaningful comparison with these existing approaches.

[1] Lusch, Bethany, et al. "Deep learning for universal linear embeddings of nonlinear dynamics." Nature communications 2018.

[2] Wang, Rui, et al. "Koopman neural forecaster for time series with temporal distribution shifts." arXiv preprint arXiv:2210.03675.

[3] Liu, Yong, et al. "Koopa: Learning Non-stationary Time Series Dynamics with Koopman Predictors." Neurips 2023.

[4] Azencot, Omri, et al. "Forecasting sequential data using consistent Koopman autoencoders." ICML 2020.

**Questions:**

None

---

### Official Review · Reviewer_t2ZP · 2023-10-26

**Soundness:** 2 fair
**Presentation:** 1 poor
**Contribution:** 2 fair
**Rating:** 5
**Confidence:** 4

**Summary:**

To address the issue of temporal distribution shift, this paper employs Koopman theory in its time-series analysis. The paper theoretically demonstrates that infinite Koopman operators can be approximated through finite matrix multiplications. Furthermore, it establishes that meta-Koopman operators are equivalent to a specific set of matrices. The model is implemented within an autoencoder framework. A comprehensive set of experiments is also conducted to validate the effectiveness of the proposed method.

**Strengths:**

1. **Theoretical Analysis**: This paper integrates meta-Koopman analysis into a time-series forecasting model, effectively transforming the data into a linear space to address domain shift issues.

3. **Proposal of Meta Koopman Operator**: The introduction of the Meta Koopman operator in this paper offers a novel approach, distinguishing it from the MMOE structure utilized in reference [1].

4. **Ease of Understanding**: The paper presents a focused effort to resolve the time domain shift problem, an aspect not extensively covered in prior research, in an easily comprehensible manner.

6. **Relevance to Conference Theme**: The paper is well-aligned with the conference topic, as it incorporates the use of a transformer model in conjunction with the Koopman operator.
----
References:
[1] Zhou, Tian, et al. "Fedformer: Frequency enhanced decomposed transformer for long-term series forecasting." International Conference on Machine Learning. PMLR, 2022.
[2]. Zhou, Tian, et al. "Film: Frequency improved legendre memory model for long-term time series forecasting." Advances in Neural Information Processing Systems 35 (2022): 12677-12690.

**Weaknesses:**

1. **Basic Conceptual Concerns**: To the best of my understanding, the paper omits the usage of 'orthogonal' or 'sparse constraints' in Section 3.2, opting instead to maintain a set of learnable meta-Koopman operators. According to Reference [1], orthogonality appears to be crucial for the bagging strategy employed.

2. **Insufficient Baseline Comparison**: The paper would benefit from presenting experimental results against more state-of-the-art time-series forecasting models, such as 'DLinear' and 'Koopman Transformer.' Additionally, the absence of comparisons with similar structures in References [2] and [3] diminishes the paper's novelty. Besides, reference [4] has mentioned that Linear structure can also perform well, why this basline is not considered?

3. **Lack of Related Works Section**: The manuscript lacks a 'Related Works' section, which could serve to identify technical gaps between the proposed work and existing literature.

4. **Figure 1 Concerns**: Figure 1 is inadequately constructed, and there is a lack of accompanying explanation. Furthermore, the bottom-right portion of Figure 1 appears to be missing from the figure.
---
References:
[1] Zhou, Tian, et al. "Fedformer: Frequency enhanced decomposed transformer for long-term series forecasting." International Conference on Machine Learning. PMLR, 2022.
[2]. Wang R, Dong Y, Arik S Ö, et al. Koopman neural forecaster for time series with temporal distribution shifts, ICLR 2023
[3]. Liu Y, Li C, Wang J, et al. Koopa: Learning Non-stationary Time Series Dynamics with Koopman Predictors, NeurIPS 2024.
[4]. Zeng, Ailing, et al. "Are transformers effective for time series forecasting?." AAAI 2023.

**Questions:**

In light of the identified weaknesses, I am inclined to assign a lower score at this time. Should the authors adequately address or convincingly rebut the weakness 1) and 3), I would consider revising my score upwards.

---

### Official Review · Reviewer_qB8r · 2023-11-04

**Soundness:** 3 good
**Presentation:** 2 fair
**Contribution:** 3 good
**Rating:** 5
**Confidence:** 4

**Summary:**

This paper presents a time series forecasting model that combines Koopman theory to tackle temporal distribution shifts. The model utilizes an auto-encoder architecture, where the encoder maps time series data into measurements, allowing for the modeling of time series dynamics using Koopman operators. The decoder then generates predictions for future states based on these estimated measurements. To handle different temporal distributions, the model includes a mechanism for matching meta Koopman operators to approximate the Koopman operator for diverse time series dynamics.

**Strengths:**

- The paper introduces a novel approach, and the core idea of the paper, particularly addressing different temporal distributions through a linear combination of meta Koopman operators, is very interesting.

-  The paper is well-written.

**Weaknesses:**

- When it comes to a finite-dimensional approximation of the Koopman operator and its set of eigenfunctions, I'm unsure about the advantages of the proposed method in comparison to other existing methods like EDMD (see the following references).

[1] Matthew O. Williams, Ioannis G. Kevrekidis, and Clarence W. Rowley, A data-driven approximation of the koopman operator: Extending dynamic mode decomposition, Journal of Nonlinear Science, 25(6):1307-1346, (2015).

[2] Christof Schütte, Péter Koltai, and Stefan Klus, On the numerical approximation of the perronfrobenius and koopman operator, Journal of Computational Dynamics, 3(1):1-12, (2016).

How does the proposed framework compare to other existing approaches in terms of accuracy and computational efficiency?

- Regarding the number of learnable meta Koopman operators (k), it is unclear how to choose the most optimal value for k. Additionally, it's uncertain how this number may impact the results.

- Limited discussion of hyperparameters: For instance, regarding T in eq. (8), it's important to understand how variations in $d_t$ can influence the results. Also, does the lack of uniqueness in T pose any potential issues?

- How does the choice of activation function affect the performance of the MLP? Were other activation functions considered, and if so, how did they compare to the chosen function?

- Can the MLP be replaced with other types of neural networks, such as convolutional neural networks or recurrent neural networks? How would this affect the performance of the proposed framework?

I am happy to increase my score if the authors could address my concerns.

**Questions:**

1- How is the proposed framework designed to address non-stationary time series, and what are some potential constraints or drawbacks of this approach within this context?

2- Since the Koopman operator for chaotic systems does not have a pure point spectrum (eigenvalues) but instead has a continuous spectrum, how can this method handle chaotic data?